# A Novel Deep Learning Model Compression Algorithm

**Ming Zhao [1][ID], Meng Li [1], Sheng-Lung Peng [2] and Jie Li [1,*]**

1   School of Computer Science, Yangtze University, Jingzhou 434025, China; hitmzhao@gmail.com (M.Z.);
    limengdas@163.com (M.L.)
2   Department of Creative Technologies and Product Design, National Taipei University of Business,
    Taipei 10051, Taiwan; slpeng@ntub.edu.tw
*   Correspondence: tntlijie@gmail.com

**Abstract:** In order to solve the problem of large model computing power consumption, this paper proposes a novel model compression algorithm. Firstly, this paper proposes an interpretable weight allocation method for the loss between a student network (a network model with poor performance), a teacher network (a network model with better performance) and real label. Then, different from the previous simple pruning and fine-tuning, this paper performs knowledge distillation on the pruned model, and quantifies the residual weights of the distilled model. The above operations can further reduce the model size and calculation cost while maintaining the model accuracy. The experimental results show that the weight allocation method proposed in this paper can allocate a relatively appropriate weight to the teacher network and real tags. On the cifar-10 dataset, the pruning method combining knowledge distillation and quantization can reduce the memory size of resnet32 network model from 3726 KB to 1842 KB, and the accuracy can be kept at 93.28%, higher than the original model. Compared with similar pruning algorithms, the model accuracy and operation speed are greatly improved.

**Keywords:** deep learning; model compression; knowledge distillation; quantization; network pruning

## 1. Introduction

As early as the mid-19th century, Warren McCulloch et al. [1] put forward the concept of the artificial neural network. However, limited by the hardware conditions at that time, the neural network did not develop well. In the 20th century, with the improvement of computing power, Hinton et al. [2] proposed a deep learning method and a new effective neural network training mechanism. In 2012, the deep learning network model AlexNet [3] built by Krizhevsky et al. won the championship in the image visual recognition competition, which was 10% more accurate than the second-ranked traditional recognition method. Since then, the convolutional neural network (CNN) has received extensive attention, and the deep learning model has been widely used in various fields, such as target tracking, pattern recognition, natural language processing and so on.

The research trend in the early stages of deep learning was to design more complex networks with more layers, from LeNet [4], AlexNet [3], VGGNet [5], GoogleNet [6] to ResNet [7], DenseNet [8] and so on. Now, the latest version of the popular Swin Transformer [9] has even reached 3 billion parameters. (Table 1 gives some basic information of classical convolutional neural networks). Generally, the performance of the model is proportional to the complexity of the model structure. However, the parameters and computation of the complex model are huge, and it is difficult to deploy to mobile embedded devices. For example, the WeChat applet requires developers to publish applications with a size of no more than 2 MB. Therefore, model compression technology has also become a hot research field. The existing model compression methods can be roughly divided into the following six categories: network pruning, quantization, knowledge distillation, network

decomposition, parameter sharing and compact network design. The main work of this paper is to combine the first three compression technologies on the basis of pruning.

**Table 1.** Basic information of some classical convolutional networks.

| Network Name | Number of Network Layers | Parameters (M) |
|---|---|---|
| AlexNet | 8 | 61.1 |
| Vgg16 | 16 | 138.36 |
| Vgg19 | 19 | 143.67 |
| GoogleNet | 22 | 6.62 |
| ResNet50 | 50 | 25.58 |

In 1989, LeCun et al. [10] proposed the idea of weight pruning, which treats all the weight parameters in the network as optimal brain damage (OBD) of a single parameter and removes the unimportant weights in the network to improve the accuracy and generalization ability of the network; Anwar et al. [11] pioneered the concept of structured pruning, using evolutionary particle filters to determine the importance of network connections; Li et al. [12] proposed a compression technique based on convolutional kernel pruning, which prunes the convolutional kernels that have less impact on the network output accuracy, reducing the computational effort by removing those convolutional kernels in the network and the feature maps to which they are connected; Chin et al. [13] use l_2-norm to measure the importance of filter, and the standardization mode of all layers was set as linear transformation. The parameter value of linear transformation of each layer was solved by evolutionary algorithm, so as to rank the importance globally. Hu et al. [14] proposed network trimming, which iteratively optimizes the network by analyzing the output of neurons on a large dataset to reduce unimportant parts. In addition to compressing classic convolutional neural networks, there are also people compressing thin networks like MobileNets [15]. The parameters of neural networks are usually 32-bit floating-point type, and the number of bits of the parameters will affect the accuracy of the model to a certain extent. Therefore, in the field of quantization, when dealing with some complex networks, the mild low-precision quantization method is generally adopted, and now the popular research direction is 8-bit fixed-point quantization [16–19]. In addition, Han et al. [20] added Huffman coding after quantization, which can further reduce the memory size and operation time of the model. Wu, Stewart and Wang et al. [21–23] designed a new quantization framework for the hardware level, and provided different quantization strategies for different neural networks and hardware structures. Besides pruning and quantization, knowledge distillation is also an effective method of model compression. It does not need to destroy the network structure and change the parameter types, but uses a simplified network to learn the knowledge in complex networks, so that it has better performance. At the end of the 20th century, Hansen and Krogh et al. [24,25] proposed the use of cross-validation to optimize the network parameters, which can effectively reduce the generalization errors in the training process. In this century, Hinton et al. [26] elaborated the dark knowledge in detail. They made the student network learn the dark knowledge and the real label in the teacher network at the same time in the training process, so that the performance of the student network was close to that of the teacher network. Subsequently, self-supervised knowledge distillation [27] and multi-teacher network knowledge distillation [28] emerged to improve the model effect of the student network from many different aspects.

The pruning methods mentioned above can reduce the amount of parameters and calculations of the model to a certain extent. However, when pruning a large proportion of complex networks, even with fine tuning, most of the pruned models will lose accuracy, to a certain extent, compared with the original models. Based on this, this paper proposes a filter pruning algorithm combining knowledge distillation and quantization. Introducing the knowledge distillation technology, while pruning the network by filter, allowed the teacher network to guide the pruned student network to train and learn the probability distribution of the output of the softmax layer of the teacher network, so as to improve the

precision of the pruned model. Finally, by quantifying the weights of the pruned model, and maintaining the precision with fine adjustment, the model size and calculation amount could be further reduced. The experimental results show that, compared with similar pruning algorithms, the model size and computational complexity of the proposed method are smaller than those of other similar algorithms under the same pruning ratio. Under the conditions of a larger pruning ratio, the accuracy can still exceed that of the original model.

The rest of this paper is arranged as follows:

Section 2 is the related foundations of knowledge distillation, quantization and pruning are introduced. In Section 3, we describe the steps and details of the algorithm proposed in this paper. In Section 4, we analyze the experimental results and deploy the model. Section 5 is the conclusion and future work.

## 2. Related Work

Before describing the algorithms proposed in this article, we will introduce some basics of knowledge distillation, quantization, and pruning in this section for further discussion.

### 2.1. Knowledge Distillation (KD)

To a certain extent, the more complex a network is, the better it performs. Knowledge distillation is to use the "knowledge" learned by a teacher's network to guide a student's network learning, so as to improve the performance of the student's network. The "knowledge" here refers to the parameters in the network. Considering the network model as a black box, the "knowledge" can be seen as the mapping relationship between the input and the input of the model. Therefore, the entire process of knowledge distillation is to first train a teacher network, and then use the output $q$ of the teacher network as the target of the student network to train the student network so that the result $p$ of the student network is close to $q$. At this time, the loss function of the student network is:

$$L = \alpha CE(y, p) + \beta CE(q, p) \tag{1}$$

The loss function of the training network without knowledge distillation is:

$$L = CE(y, p) \tag{2}$$

Here, $CE$ stands for cross entropy, $y$ is the one-hot code of real label, $q$ is the output result of teacher network, $p$ is the output result of student network, and $\alpha$ and $\beta$ are hyperparameters, which are used to control the weights of the two cross entropies. Figure 1 shows the flow chart of knowledge distillation.

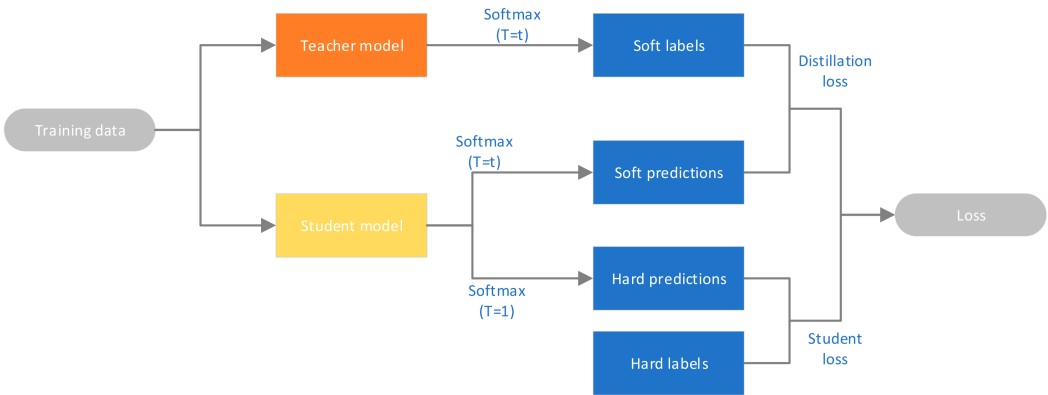

**Figure 1.** Flow chart of knowledge distillation.

However, if the output $q$ of the softmax layer of the teacher network is directly used to guide the student network, there will be some problems. This is because a network, after being trained, will have a high confidence level for the correct answer. For example, if you

input a picture of a deer, the network's prediction probability for the deer will be very high, but for a category similar to the deer, such as a horse, the prediction probability is very low. In this case, the similar information between data learned by the teacher network (such as the similar information between deer and horse) is difficult to convey to the student network. Therefore, Hinton proposed softmax-*T*, with the following equation:

$$q_i = \frac{\exp(z_i/T)}{\sum_j \exp(z_i/T)} \tag{3}$$

Here is the object learned by the student network (soft targets) and is the output logits of the teacher network before softmax. If *T* is taken as 1, this formula is the original softmax, which outputs the probability of each category according to logits:

$$\text{Softmax}(z_i) = \frac{e^{z_i}}{\sum_{c=1}^{C} e^{z_c}} \tag{4}$$

where is the output value of the *i*-th node and *C* is the number of output nodes, i.e., the number of categories of the classification. Through the softmax function, the output value of the multi-classification can be converted into a probability distribution in the range of [0, 1].

If t is close to 0, the maximum value will be closer to 1, and other values will be close to 0, which is similar to one-hot coding. If t is larger, the distribution of the output results will be smoother, which is equivalent to smoothing and plays a role of retaining similar information. If t equals infinity, it is a uniform distribution. Figure 2 is the probability distribution diagram of softmax with different T values on the handwritten digital dataset MNIST. We can see that the softened probability distribution can reflect that this handwritten digital picture has a certain degree of similarity between 2 and 8.

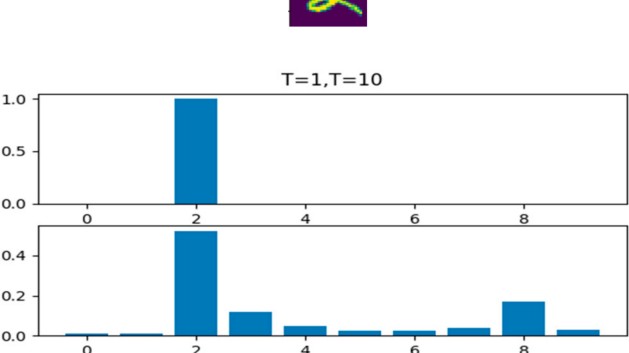

**Figure 2.** Probability distribution of different T values.

### 2.2. Quantization

Quantization in the traditional sense refers to discretizing continuous signals [29], for example, changing people's continuous sound signals into discrete digital signals. Taking weight quantization as an example, the quantization in neural network refers to changing the weights stored in the form of float32 into int8, int4, int2 and other types, reducing the size of their representable space, and establishing the corresponding mapping relationship, as shown in Figure 3. The purpose of neural network quantization is also to reduce the space occupied by the weights of neural networks. However, unlike pruning, quantification solves the problem in the storage form of parameters, while pruning directly removes redundant connections in the network.

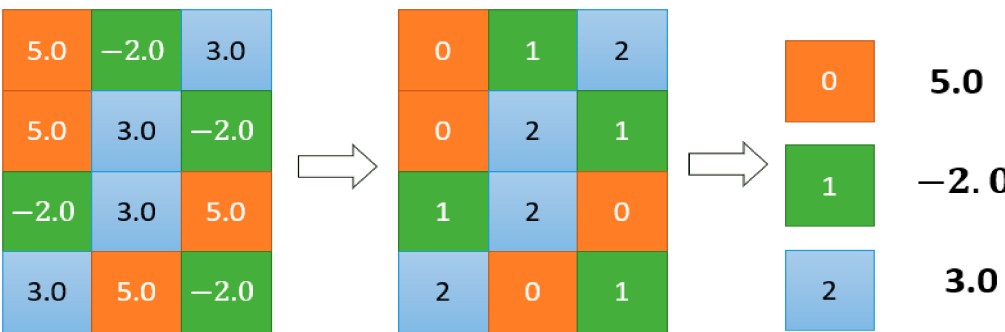

**Figure 3.** Schematic diagram of weight quantification.

Model quantization is mainly divided into two steps: quantization and inverse quantization. The whole process is to first input the float-type weights in the model, obtain the maximum and minimum values of the weights, quantify the weights according to them, and then retrain until the network converges. At the same time as training, finding the corresponding relationship to inverse quantize the output values to obtain the final result. The whole process is shown in Figure 4.

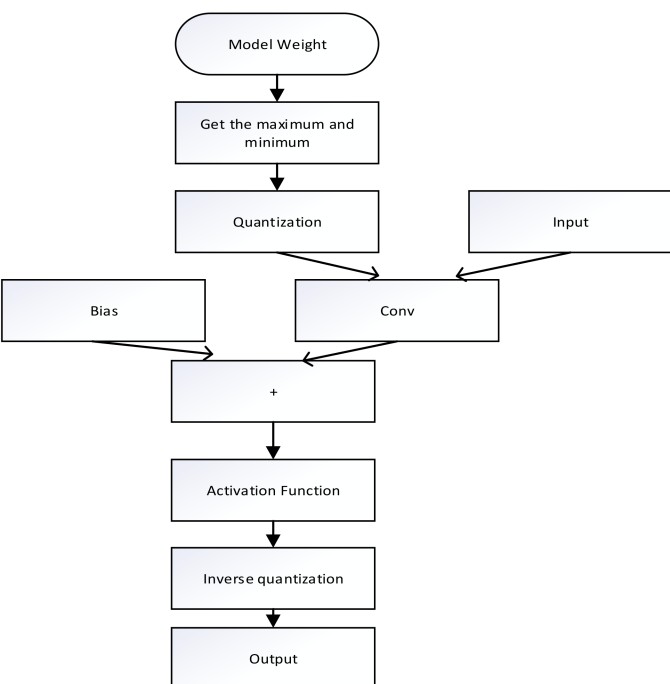

**Figure 4.** Flow chart of model quantification.

### 2.3. Pruning

Pruning is seeking a judging mechanism in the process of network training, to eliminate unimportant connections, nodes and even convolution kernels in the network in order to achieve the purpose of simplifying the network structure. Many experiments have shown that there are a lot of redundant parameters from the convolutional layer to the fully connected layer, and even if these neurons are eliminated, the model characteristics can be expressed [30].

Pruning methods can be divided into two categories: structured pruning and unstructured pruning. Earlier research methods were mainly based on unstructured, which prunes at the granularity of a single neuron. If the convolutional kernel is unstructured pruned, the resulting convolutional kernel is sparse, i.e., it has many matrices with element 0 in the middle. Unless the lower-level hardware and computational libraries support it relatively well, it is difficult to obtain substantial performance gains in the pruned model. Therefore,

much of the research in recent years has been focused on structured pruning. Structured pruning is the operation on the channel or the whole filter, which does not destroy the original convolution structure, is compatible with the existing hardware and library, and is more suitable for deployment on hardware.

The pruned model has the following three advantages:

(1) Less training time. With less computation, the speed of each iteration of connections in the network is improved and the network model can converge to the optimal solution faster.
(2) Faster operation. The sparse network has fewer convolutional layers and fewer convolutional kernels in the convolutional layers, and the simpler and lighter model means more efficient and fast weight updates.
(3) More viable embedded deployments. The pruned network offers a wider range of possibilities for applications on mobile and other embedded devices.

### 3. Fusion Pruning Algorithm

In Section 2, we briefly introduce three commonly used model compression methods, each of which has its advantages and disadvantages and adaptation scenarios. In this section, we organically combine these three pruning methods to obtain a fusion pruning algorithm, and propose an interpretable weight distribution method for knowledge distillation. Below we will introduce the algorithm steps in detail.

The algorithm proposed in this paper is mainly divided into two stages—the pruning stage and the quantization stage. In the pruning stage, we introduce knowledge distillation to guide the training model and improve the accuracy of the model. Then, the weights of the pruned model are quantified to further reduce the model size and computational power consumption. The whole flow chart is shown in Figure 5.

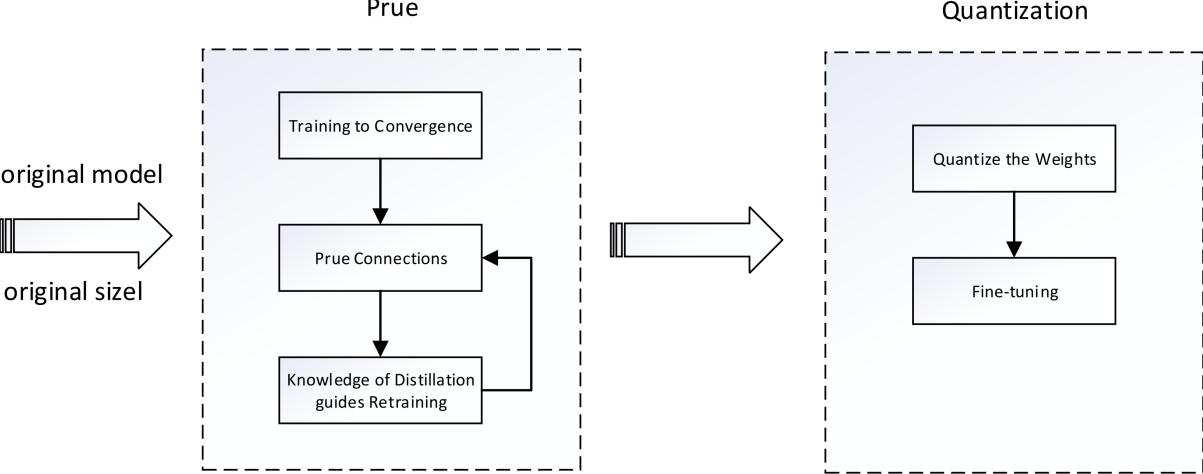

**Figure 5.** Flow chart of the algorithm proposed in this paper.

The pruning method used in this paper is filter pruning via geometric median (FPGM) proposed by He et al. [31]. Previous filter pruning methods assume that the smaller the norm of the filter, the smaller its contribution to the network [12,32]. Removing those filters with a norm close to zero would not seriously affect the performance of the network. However, this approach actually contains two implicit conditions:

(1) The standard deviation of the norm is sufficiently large.
(2) The smallest norm is close to zero.

However, in the actual pruning process, the above conditions are not always met, so this paper adopts FPGM which does not need these two conditions. The geometric median

is defined as follows: given a set of points $a^{(1)}, \ldots, a^{(n)}$, where each point $a^{(i)} \in \mathbb{R}^d$, find a point $x^* \in \mathbb{R}^d$ such that the sum of the Euclidean distances to them is minimized.

$$x^* = \arg\min_{x \in \mathbb{R}^d} f(x) \text{ where } f(x) \stackrel{\text{def}}{=} \sum_{i \in [1,n]} \|x - a^{(i)}\|_2 \tag{5}$$

The geometric median is an estimate of the center of a point in Euclidean space, and FPGM considers filters as points in Euclidean space, so we can compute the GM to get the "center" of these filters, i.e., their common properties. If a filter is close to this GM, it can be assumed that the information of this filter overlaps with other filters, or is even redundant. After removing it, its function can be replaced by other filters. That is, such filters satisfying Equation (6).

$$\begin{aligned}\mathcal{F}_{i,x^*} &= \arg\min_{x} \sum_{j' \in [1,N_{i+1}]} \| x - \mathcal{F}'_{i,j} \|_2, \text{ s.t. } x \in \{\mathcal{F}_{i,1}, \ldots, \mathcal{F}_{i,N_{i+1}}\} \\ &\stackrel{\text{def}}{=} \arg\min_{x} g(x), \text{s.t. } x \in \{\mathcal{F}_{i,1}, \ldots, \mathcal{F}_{i,N_{i+1}}\}\end{aligned} \tag{6}$$

The difference between pruning based on geometric median and pruning based on norm size is shown in Figure 6. After pruning the model, the teacher network is used to guide the pruned model, so that it can learn the knowledge of the teacher network, and make the final output result of the softmax layer of the network approach that of the teacher network, thus adjusting the weight of the whole network. In the process of knowledge distillation, it is necessary to assign a weight to the teacher network and the real label. Experience shows that the greater the weight assigned to the teacher network in the process of knowledge distillation, the closer the effect of the student network is to the teacher network. However, the accuracy rate of the teacher network is not 100%. In other words, if it is given a greater weight, the student network will also learn most of the wrong knowledge when learning the knowledge of the teacher network. Therefore, this paper proposes a new method to allocate the weights of teacher network and student network. First, identify the teacher network $A$ and student network $B$. Then both networks are used as teacher networks to perform knowledge distillation on another student network $C$. Obtain the promotion $I_A$ and $I_B$ of $A$ and $B$ to $C$ respectively. Finally, according to formula (7), assign corresponding weights $\alpha$ and $\beta$ to $A$ and $B$.

$$\alpha = \frac{I_A}{I_A + I_B}, \beta = \frac{I_B}{I_A + I_B} \tag{7}$$

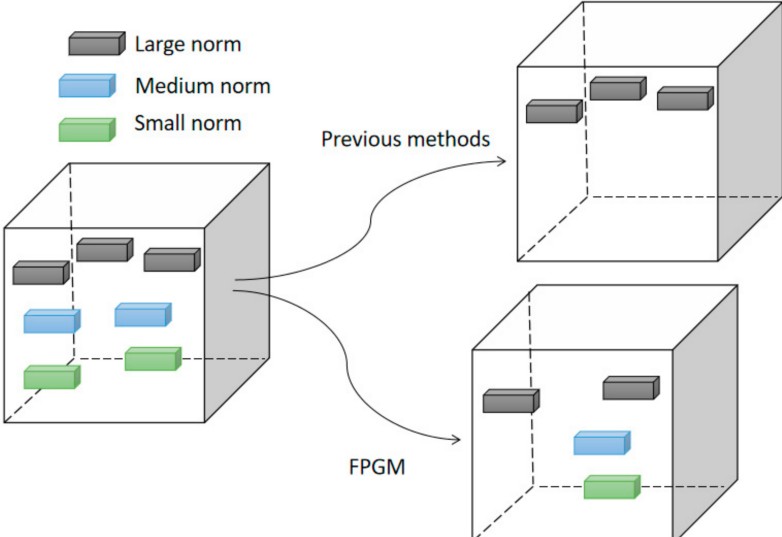

**Figure 6.** Comparison of different pruning methods.

Finally, we quantify the residual weights of the model after pruning and knowledge distillation. Quantification is described in mathematical language, that is, when two matrices $X$ and $Y$ are operated (as shown in Figure 7), $X$ and $Y$ are quantized into $X_Q$ and $Y_Q$ respectively. Among them

$$X \times Y = Z \tag{8}$$

$$X = X_Q \times a + b \tag{9}$$

$$Y = Y_Q \times c + d \tag{10}$$

$$\begin{aligned} Z &= (X_Q \times a + b) \times (Y_Q \times c + d) \\ &= ac X_Q Y_Q + ad X_Q + bc Y_Q + bd \end{aligned} \tag{11}$$

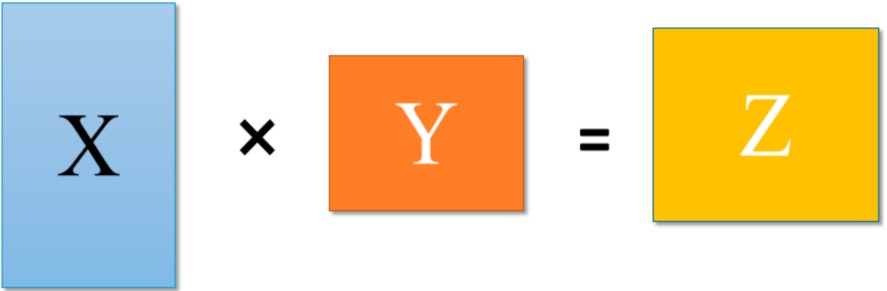

**Figure 7.** Matrix operation diagram.

The whole process is to change the matrices $X$ and $Y$ whose original data type is float32 into $X_Q$ and $Y_Q$ whose data type is int8, so as to reduce the memory size and calculation time.

## 4. Experiment and Model Deployment

### 4.1. Experiment

In order to verify the effect of the fusion pruning algorithm proposed in this paper, we validated it on the cifar-10 dataset. The cifar-10 dataset contains 60,000 32 × 32 color images in 10 different categories, including 50,000 training images and 10,000 test images. Figure 8 is a partial picture of cifar-10. The experimental equipment was a PC with windows system, the cpu was intel i5-6400, the Gpu was GTX960M and the running memory was 12 GB.

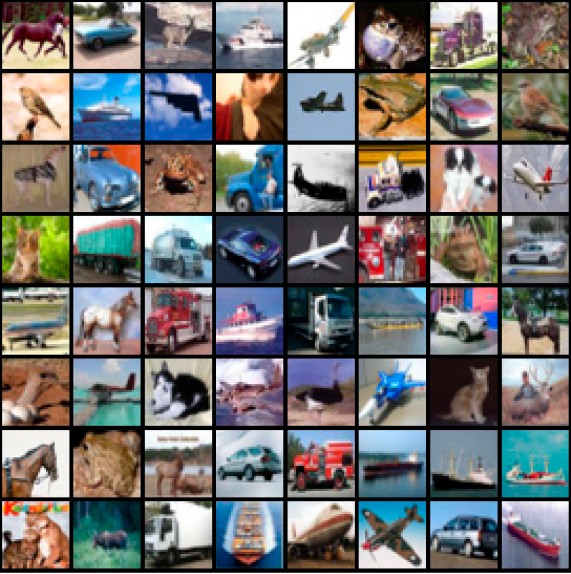

**Figure 8.** Partial picture of cifar10 dataset.

Taking the fusion and pruning of the resnet32 network as an example: First, we used the resnet32 model and densenet100 model to perform knowledge distillation on an 18-layer network handwritten by ourselves under the same temperature and weight (here we still set a larger weight for the teacher), the improvement is $I_A = 1.46\%$, $I_B = 3.41\%$, and according to the idea of determining weights in Section 3, $\alpha = I_A/(I_A + I_B) = 0.29$, $\beta = I_B/(I_A + I_B) = 0.71$. According to the data in Table 2, it can be seen that the assigned weight was very appropriate. Figure 9 shows the graph of the change in accuracy of the DenseNet guiding the ResNet training process.

**Table 2.** The effect of DenseNet_kd_ResNet under different weights.

| Weight Distribution ($\alpha$, $\beta$) | Accuracy after Knowledge Distillation |
|---|---|
| $\alpha = 1$, $\beta = 0$ (baseline) | 92% |
| $\alpha = 0.9$, $\beta = 0.1$ | 91.32% |
| $\alpha = 0.8$, $\beta = 0.2$ | 91.74% |
| $\alpha = 0.7$, $\beta = 0.3$ | 92.68% |
| $\alpha = 0.6$, $\beta = 0.4$ | 93.07% |
| $\alpha = 0.5$, $\beta = 0.5$ | 92.89% |
| $\alpha = 0.4$, $\beta = 0.6$ | 92.57% |
| $\alpha = 0.3$, $\beta = 0.7$ | 93.74% |
| $\alpha = 0.2$, $\beta = 0.8$ | 93.78% |
| $\alpha = 0.1$, $\beta = 0.9$ | 93.27% |

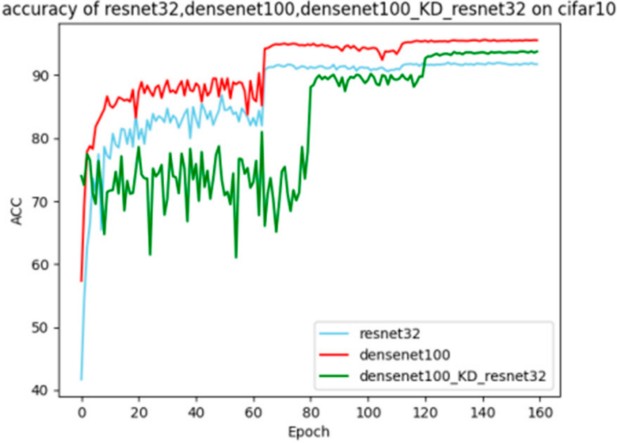

**Figure 9.** Knowledge distillation training process.

After obtaining the pruning model after knowledge distillation, we then quantified it, converted the remaining weight of the model from float32 type to int8 type, and fine-tuned it. In this paper, experiments were carried out on VGG16 and Resnet32, and compared with other methods. Finally, the comparison of models under different pruning ratios is shown in the table below.

In the end, the model size of Vgg16 changed from 115 MB to 45 MB, and the model size of resnet32 changed from 3726 KB to 1842 KB. The experimental results (Table 3) show that the pruning method proposed in this paper, which combines knowledge distillation and quantification, is superior to the single pruning method in model accuracy and model size reduction under the same pruning rate. Moreover, under the conditions of a large pruning rate, the accuracy can exceed the original model.

**Table 3.** Comparison results of different pruning methods on the cifar-10 dataset.

| Model Name | Method | Accuracy | Flops |
|---|---|---|---|
| Vgg16 (Pruning rate 50%) | Baseline | 93.14% | $6.26 \times 10^8$ |
| | PF [12] | 93.27% | |
| | HRank [33] | 92.47% | $2.31 \times 10^8$ |
| | Ours | 93.86% | |
| Resnet32 (Pruning rate 50%) | Baseline | 91.97% | $6.88 \times 10^7$ |
| | PF [12] | 92.54% | |
| | HRank [33] | 92.67% | $2.49 \times 10^7$ |
| | Ours | 93.28% | |

*4.2. Model Deployment*

The ultimate goal of studying model compression techniques is to deploy the model on mobile embedded devices. Therefore, in this part, we deployed the model compressed by the algorithm proposed in this paper on the mobile side. Android Studio was used to simulate this deployment. The simulated mobile phone environment was Android 11.0, and the mobile phone model was Pixel 4. According to the flow chart shown in Figure 10, the Pytorch model was first converted into ONNX intermediate format and then into NCNN model. Then the model was deployed to APP with the help of NCNN framework, and the final display effect is shown in Figure 11. Furthermore, we compared the time required for the original model and the pruned model to process the same input image, and the results are shown in Table 4.

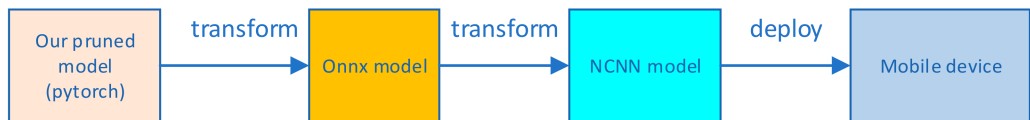

**Figure 10.** Flow chart of model deployment.

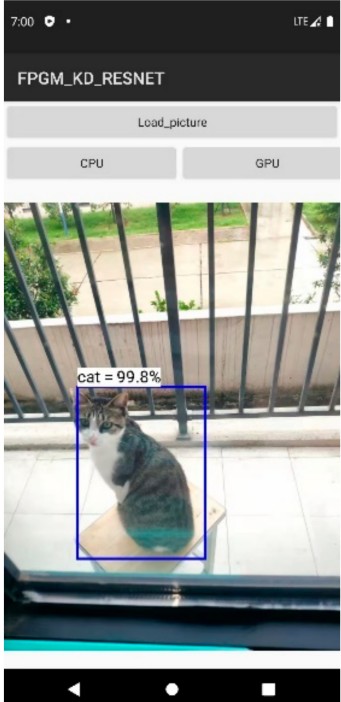

**Figure 11.** Final display effect.

**Table 4.** Comparison of the time consumption of the model before and after pruning.

| Model | Time |
|---|---|
| Original model (Vgg16) | 867 ms |
| Pruned model | 472 ms |

## 5. Conclusions and Future Work

Different from the previous pruning methods, this paper introduces knowledge distillation and weight quantization into the pruning process, and proposes an interpretable weight allocation method for the problem of weight allocation in the knowledge distillation process. Experiments show that, compared with the previous pruning methods, the method proposed in this paper can maintain or even improve the model accuracy while pruning on a large scale. Of course, this work also has certain limitations. Since this paper introduces the guidance of knowledge distillation in the pruning process, in order to facilitate processing, the fully connected layer of the model was not pruned; in addition, limited by the experimental conditions, the algorithm in this paper has not been verified on a larger dataset. In addition, in the future, more advanced knowledge distillation, pruning and quantification methods [34–36] can be used to further improve the model performance.

**Author Contributions:** Validation, writing original draft, M.Z.; Investigation, M.L.; Methodology, Supervision, S.-L.P.; Writing-review and editing; J.L. All authors have read and agreed to the published version of the manuscript.

**Funding:** This research was supported by Hubei Provincial Department of Education: 21D031.

**Conflicts of Interest:** The authors declare no conflict of interest.

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
