# Peer review of "A Novel Deep Learning Model Compression Algorithm"

_electronics, doi:10.3390/electronics11071066_

Round 1

Reviewer 1 Report

The paper is about A Novel Model Compression Algorithm and Its IoT Application. I have the following comments:

  1. The first sentence in the Abstract is 6 lines and this is too much. Please edit it to use short sentences
  2. The Abstract needs to be re-written. What do you mean by student network? Teacher network?
  3. The word ‘IoT’ is mentioned in the title but never mentioned in the text. How?
  4. The presentation of the paper needs a lot of improvement. For example, there should be a space before each reference
  5. What are the main contributions of the paper since this is unclear to me
  6. Related Works à Related Work
  7. In Table 2, add a column called “Accuracy before knowledge distillation” so that you can compare before and after
  8. It is important to study the complexity in terms of Big O notation or training /testing time for the models before and after pruning. This is in addition to the accuracy
  9. The case study is very short and incomplete
  10. What are the limitations of the proposed work?

Reviewer 2 Report

Dear authors, I hope that these comments will contribute to improve your article.

Abstract, check the language, for example: "In order to solve the problems that deep learning model consumes a lot of computing power, runs".

Part of the introduction could move to literature review.
In the introduction it is not clear what is new with respect to the previous one.

The literature review lacks bibliographic references. 

It is recommended to the authors to revise the document since most of it looks more like a handbook than a scientific article. 

Conclusions should be specified with which previous algorithms.

The relevance of the work with respect to the previous ones is not clear. What is really new and remarkable? What does the work contribute to the literature?

The applications to the IoT are not shown neither in the objectives nor in the conclusions, although they are mentioned in the title.

I would recommend reviewing the limitations, since their reading leads to the conclusion that the work is not relevant.

Round 2

Reviewer 1 Report

The authors addressed my comments

Author Response

Thank you for your previous corrections and suggestions, thank you very much.

Reviewer 2 Report

Dear authors, congratulations!!

Author Response

Thanks for your valuable comments.